# Comparison of Clinical Characteristics between Responders and Non-Responders to Hamstring Stretching in Individuals with Chronic Low Back Pain

**DOI:** 10.3390/diagnostics14192117

**Published:** 2024-09-24

**Authors:** Lech Dobija, Bruno Pereira, Arnaud Dupeyron, Emmanuel Coudeyre

**Affiliations:** 1Service de Médecine Physique et de Réadaptation, Centre Hospitalier Universitaire (CHU) de Clermont Ferrand, 63000 Clermont-Ferrand, France; 2Unité Nutrition Humaine, Institut National de Recherche pour l’Agriculture, l’Alimentation et l’Environnement (INRAE), Université Clermont Auvergne, 63000 Clermont-Ferrand, France; 3Direction de la Recherche Clinique et de l’Innovation, Centre Hospitalier Universitaire (CHU) de Clermont-Ferrand, 63000 Clermont-Ferrand, France; 4Service de Médecine Physique et de Réadaptation, Centre Hospitalier Universitaire (CHU) de Nîmes, 30900 Nîmes, France

**Keywords:** hamstring muscle, chronic low back pain, stretching

## Abstract

**Background/Objectives:** Hamstring muscle (HM) flexibility is frequently compromised in people with chronic low back pain (CLBP), contributing to disability and leading to a less favorable recovery. In a previous article, we presented the results of a study on the immediate effect of passive HM stretching on flexibility in 90 people with CLBP. There was considerable variability in the changes after stretching. The objective of this supplementary analysis was to compare the clinical characteristics of individuals who experienced a significant improvement in flexibility with those who did not. **Methods:** We fixed a threshold of 7° to indicate an improvement in passive Straight Leg Raise (SLR) angle and differentiate between ‘Responders’ and ‘Non-responders’ regarding passive HM stretching. **Results:** Only body mass index differed between groups; it was significantly smaller in Responders (*p* = 0.007). The majority of Non-responders experienced workplace accidents, but this was of marginal difference compared to Responders (*p* = 0.056). **Conclusions:** Further studies should consider a broader clinical analysis with a larger sample size to understand which factors influence the HM stretching response in CLBP patients.

## 1. Introduction

Hamstring muscle (HM) flexibility is frequently compromised in people with chronic low back pain (CLBP), contributing to disability and leading to a less favorable recovery [1,2,3]. In a previous article, we presented the results of a study on the immediate effect of passive HM stretching on flexibility in 90 people with CLBP [4]. The improvement in HM flexibility following one-minute passive stretching was statistically significant, with the Straight Leg Raise (SLR) angle showing a mean improvement of 7° (95% CI 5.5 to 8.6°, *p* < 0.001, ES: 0.42–0.44), Active Knee Extension angle showing a mean improvement of 4° (95% CI 2.4 to 5.1°, *p* < 0.001, ES: 0.23–0.24) and Fingertip-to-Floor distance showing a mean improvement of 2 cm, (95% CI 1.7 to 3.0 cm, *p* < 0.001, ES = 0.20). Furthermore, there was considerable variability in the changes after stretching; substantial improvements occurred in some individuals, whereas others exhibited minimal or no change. From a clinical perspective, understanding the reasons behind the different responses to passive HM stretching in people with CLBP is very important. The objective of this supplementary analysis was to compare the clinical characteristics of individuals who experienced a significant improvement in flexibility with those who did not.

## 2. Materials and Methods

The original study was approved by a local ethics committee on 10 February 2020 (Comité de Protection des Personnes—Ouest 1, Identifier: 2020T2-01_RIPH2 HPS_2019-A03000-57) and informed consent was obtained from all participants. A detailed description of the study methods, measurement, and intervention procedure was presented in a previous article [4]. Briefly, the SLR angle was measured with a digital inclinometer before and immediately after a one-minute session of passive hamstring stretching. Clinical characteristics (e.g., age, sex, body mass index, Oswestry Disability Index, Fear-Avoidance Beliefs Questionnaire, and Hospital Anxiety and Depression Scale) were recorded at baseline. The participants’ ages ranged from 18 to 60 years. In the present supplementary analysis, we fixed a threshold of 7° to indicate an improvement in passive SLR angle and differentiate between ‘Responders’ and ‘Non-responders’ regarding passive hamstring stretching. This threshold was set based on the mean improvement in SLR angle of the less flexible lower limb, which was 7°, and the previously estimated Minimal Detectable Change, which ranged from 6.8° to 10.4° [4]. Additionally, we performed a correlation analysis between clinical characteristics and the improvement in the passive SLR angle in Responders, Non-responders, and all participants.

Continuous data were expressed as mean and standard deviation. The assumption of normality of the distribution was analyzed using the Shapiro–Wilk test. Comparisons between groups (Responders vs. Non-responders) were performed using the chi-squared or Fisher’s exact tests for categorical data. Student’s *t*-test or non-parametric Mann–Whitney test (when the assumptions of the t-test were met) were used for the comparisons concerning continuous variables. Relationships between continuous data were analyzed using Pearson or Spearman correlation coefficients, depending on the statistical distribution.

Additionally, a sensitivity analysis was performed to verify whether using a threshold other than 7° would yield different results and to ensure the robustness of our findings. Specifically, we examined whether using a threshold between 6° and 8° for improvement in the passive SLR angle would change the comparison results of clinical characteristics between ‘Responders’ and ‘Non-responders’.

Statistical analyses were performed using Stata 15 software (StataCorp, College Station, TX, USA). All tests were two-tailed, with a type I error set at 0.05.

## 3. Results

After 1 min of passive stretching, the improvement in SLR angle was equal to or exceeded 7° in 28 participants (31%) (Responders); the change was less than 7° (Non-responders) in 62 participants (69%). We found no statistically significant differences in any of the clinical characteristics analyzed between the Responders and the Non-responders except for body mass index (BMI), which was greater in the Non-responder group. The results also showed that of the 90 participants, 17 had experienced a workplace accident. The majority of these individuals (*n* = 15, 88%) were in the Non-responder group; however, this difference was marginal (*p* = 0.056) (Table 1).

Furthermore, a moderate statistically significant correlation between BMI and the improvement in the passive SLR angle in Responders was found (r = 0.44, *p* < 0.05). There were no other statistically significant correlations (Table 2). A supplementary verification of a threshold of 6° to 8° for improvement in the passive SLR angle revealed very similar distributions between the groups. Specifically, using a 6° threshold identified 33 Responders and 57 Non-responders, while an 8° threshold identified 25 Responders and 65 Non-responders. In both cases, only the BMI differed significantly between the groups. Using a 6° threshold, the BMI was 28.1 ± 5.9 in Non-responders compared to 25.7 ± 5.2 in Responders (*p* = 0.03). Using an 8° threshold, the BMI was 27.8 ± 5.8 in Non-responders compared to 25.7 ± 5.5 in Responders (*p* = 0.05).

## 4. Discussion

We performed this supplementary analysis to identify clinical differences between Responders and Non-responders regarding passive HM stretching. However, only BMI differed between groups; it was significantly smaller in Responders. The reason for the effect of BMI on stretch efficacy is not obvious.

To our knowledge, no other study has evaluated the relationship between BMI and HM stretching effects in people with CLBP. However, reduced muscle flexibility has been associated with higher BMI in older women [5], and low back pain has been linked to both higher BMI and decreased HM flexibility [6]. Anxiety has been shown to be more frequent in people with obesity or overweight than in the general population [7]. Anxiety could potentially reduce the capacity to relax muscles during stretching and thus reduce stretching efficacy. In addition, higher levels of fear of movement have been reported among people with CLBP and obesity compared to those without obesity [8]. However, we found no correlation between Hospital Anxiety and Depression Scale (HADS) and Fear-Avoidance Beliefs Questionnaire (FABQ) scores and improvement in flexibility [4], suggesting that other factors may play a role in the BMI and stretching effect relationship. On the other hand, the application of stretching force on the HM by the therapist requires greater effort in individuals with obesity or overweight due to the increased mass of the lower limb. This, in turn, may influence the effectiveness of stretching. Yet, in the Responders, improvement in passive SLR was correlated with higher BMI, indicating that the influence of BMI on the stretching effect could be more complex. Despite a marginal difference (*p* = 0.056), the results showed that of the 90 participants, 17 had experienced a workplace accident, and the majority of these individuals (*n* = 15, 88%) were in the Non-responder group. Workplace-related factors are known to impact the recovery of patients with CLBP; therefore, a workplace accident could indeed influence the stretching effect [9]. We also noticed no significant difference regarding gender between Responders and Non-responders, indicating that being male or female is independent of the HM stretching effect.

The interpretation of this supplementary analysis is limited by the sample size; a much larger sample size would be required to draw robust conclusions for such a group-based analysis. However, we can state that varying the choice of threshold used to distinguish between the groups revealed similar results. Therefore, it should not be considered a source of bias in this study. The hypothesis of the initial study was that psychosocial factors would impact stretching efficacy [4], but the overall and group-based analyses revealed no relationship between these factors. The question of why HM flexibility improves after HM stretching in some individuals can therefore not simply be explained by psychosocial characteristics evaluated by the FABQ and HADS questionnaires. Other clinical factors like the state of lumbar degenerative changes, flexibility of other muscles (e.g., piriformis, erector spinae, and hip adductors), and neurodynamic issues need to be considered. In the presence of pain, muscles in the posterior chain may contract simultaneously during stretching [10], which may prevent effective stretching of the HM. Furthermore, the SLR does not only stretch the HM; therefore, its amplitude may be limited by other anatomical structures. One study found that increases in HM flexibility measure could be achieved through myofascial release techniques applied to the posterior muscle chain but not specifically the HM [11]. Therefore, it is important to thoroughly examine the individual to find the cause or causes of the reduced SLR angle. None of the participants presented radicular pain. Therefore, it is unlikely that the stretching effect was limited by radicular pain. However, some participants had degenerative changes within the intervertebral disc or a history of radicular pain that might have influenced neurodynamics, even in the absence of evident radicular symptoms. These conditions could potentially impact the effectiveness of stretching [12]. Future studies should take the above factors into account in order to explain why some CLBP patients improve HM flexibility while others do not. In conclusion, the only clinical factor found to relate to a positive response to HM stretching was a low BMI. This indicates that people with overweight or obesity are more likely to present an unsatisfactory response to HM stretching. Other clinical factors, such as age, gender, pain and disability levels, fear-avoidance beliefs, and symptoms of anxiety and depression, appear to have no significant impact.

## Figures and Tables

**Table 1 diagnostics-14-02117-t001:** Comparison of clinical characteristics between Responders and Non-responders.

	Responders	Non-Responders	*p* Value
Number and percentage of participants	28 31%	62 69%	
Age [years]	45.6 ± 8.9	43.8 ± 9.2	0.372
Men	16 (30%)	37 (70%)	0.822
Women	12 (32%)	25 (68%)	
BMI [kg/m^2^]	25.2 ± 5.5	28.1 ± 5.7	0.007 *
Education level:			0.152
No diploma	0 (0%)	1 (100%)	
Less than baccalaureate	13 (30%)	30 (70%)	
Baccalaureate level	8 (25%)	24 (75%)	
Higher education studies	7 (58%)	5 (42%)	
Type of work:			0.214
Sedentary	9 (43%)	12 (57%)	
Physical	11 (23%)	36 (76%)	
Mixed	7 (37%)	12 (63%)	
Living environment:			0.887
Urban	19 (31%)	43 (69%)	
Rural	9 (32%)	19 (68%)	
Active smoking			0.644
Yes	10 (27%)	27 (73%)	
No	18 (34%)	35 (66%)	
Workplace accident			0.056
Yes	2 (12%)	15 (88%)	
No	26 (36%)	47 (64%)	
Time since pain onset [months]	73.9 ± 97.5	84.1 ± 88.9	0.295
Pain before stretching			
VAS [0–100]	36.8 ± 21.8	38.2 ± 22.5	0.877
Pain after stretching			
VAS [0–100]	40.5 ± 21.6	42.7 ± 23.8	0.835
Pain change			
VAS [0–100]	3.7 ± 12.6	4.0 ± 18.1	0.605
ODI	34.4 ± 14.2	34.4 ± 12.0	0.785
FABQ Physical Activity	14.6 ± 5.4	14.2 ± 6.3	0.949
FABQ Work	25.0 ± 11.3	28.5 ± 10.6	0.193
HADS Anxiety	10.1 ± 3.8	10.1 ± 3.5	0.953
HADS Depression	7.9 ± 3.1	8.1 ± 3.3	0.728

Data are *n* (%) or mean ± SD. BMI, body mass index; VAS, Visual Analogue Scale; ODI, Oswestry Disability Index; FABQ, Fear-Avoidance Beliefs Questionnaire; HADS, Hospital Anxiety and Depression Scale; * *p* < 0.05; ‘Responders’ were defined as those with an improvement of ≥7° in Straight Leg Raise angle after stretching and ‘Non-responders’ as a change <7°.

**Table 2 diagnostics-14-02117-t002:** Correlations between improvement in passive Straight Leg Raise angle and clinical characteristics.

	All Participants (*n* = 90)	Non-Responders (*n* = 62)	Responders (*n* = 28)
Age	0.10	−0.04	0.11
BMI	−0.15	0.05	0.44 *
Time since pain onset	−0.16	−0.10	−0.13
Pain VAS before stretching	0.03	0.08	0.10
Pain VAS after stretching	0.03	0.05	0.15
Pain VAS change	0.05	0.00	−0.02
ODI	−0.13	−0.10	0.07
FABQ Physical Activity	0.07	0.10	0.08
FABQ Work	−0.07	0.16	0.31
HADS Anxiety	−0.09	−0.23	0.03
HADS Depression	−0.10	−0.14	−0.01

BMI, body mass index; ODI, Oswestry Disability Index; VAS, Visual Analogue Scale; FABQ, Fear-Avoidance Beliefs Questionnaire; HADS, Hospital Anxiety and Depression Scale; * *p* < 0.05; ‘Responders’ were defined as those with an improvement of ≥7° in Straight Leg Raise angle after stretching and ‘Non-responders’ as a change <7°.

## Data Availability

Dataset is available on request from the authors.

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
