# Peer review of "Comparison of Clinical Characteristics between Responders and Non-Responders to Hamstring Stretching in Individuals with Chronic Low Back Pain"

_diagnostics, 2024, doi:10.3390/diagnostics14192117_

Round 1

Reviewer 1 Report

Comments and Suggestions for Authors

The scientific contribution of the manuscript seems limited because it mostly repeats previous results with an extra focus on BMI. Is MBI that important? That is questionable. The study suggests that more research with larger groups and wider clinical analysis is needed. 

Minor issues:

Line 31: remove extra dot here "ery.[1–3]."

Lines 48-49: remove extra dot here "a previous article. 48 [4]."

Line 79: remove extra dot here "correlations. (Table 2.)"

Line 37: add space in "2cm" and "3cm"

Line 63: "guarantee" instead of "guaranty"

Line 71 & Table 1: should be "69%" instead of "70%", because 62/90=69%

Line 72: "analyzed" instead of "analysed"

Line 76: dot should be after ". (Table 1)"

starting from Line 95: the citations should not be in bold.

Line 120: instead of "(ex. " should be used "(e.g."

Comments on the Quality of English Language

It is Ok, but some typos should be fixed as discussed above.

Author Response

Comments 1: The scientific contribution of the manuscript seems limited because it mostly repeats previous results with an extra focus on BMI. Is MBI that important? That is questionable. The study suggests that more research with larger groups and wider clinical analysis is needed. 

Response 1: Thank you for taking the time to evaluate our manuscript and for your comments. Indeed, our results highlight the importance of BMI in the stretching effect, which was not initially suspected to be the main factor differentiating Responders from Non-responders to hamstring stretching. We discuss the role of BMI in relation to the existing literature, acknowledge the limitations of our study, and suggest directions for future research.

Comments 2: Minor issues:

Line 31: remove extra dot here "ery.[1–3]."

Lines 48-49: remove extra dot here "a previous article. 48 [4]."

Line 79: remove extra dot here "correlations. (Table 2.)"

Line 37: add space in "2cm" and "3cm"

Line 63: "guarantee" instead of "guaranty"

Line 71 & Table 1: should be "69%" instead of "70%", because 62/90=69%

Line 72: "analyzed" instead of "analysed"

Line 76: dot should be after ". (Table 1)"

starting from Line 95: the citations should not be in bold.

Line 120: instead of "(ex. " should be used "(e.g."

It is Ok, but some typos should be fixed as discussed above.

Response 2: Thank you for pointing out the typos in the text. We have now corrected all the errors you mentioned.

Reviewer 2 Report

Comments and Suggestions for Authors

It is an interesting topic.

However, there are many aspects that need to be improved

Lines 30-31: ,, recovery.[1–3].

Lines 47-49: ,,A detailed description of the study methods, measurement, and intervention procedure was presented in a previous article.  [4].

The template for placing the reference number in the text is not respected , respectively [ ].

Lines 45-47: ,,The original study was approved by a local ethics committee (Comité de Protection des Personnes – Ouest 1, Identifier: 2020T2-01_RIPH2 HPS_2019-A03000-57) and informed consent was obtained from all participants.”

What is the age group of the study participants? In the table below, you specify the average age, but what are the age limits?

Lines 47-49: ,,A detailed description of the study methods, measurement, and intervention procedure was presented in a previous article.  [4].

I think it was better to write briefly some aspects of the study you mention.

The objective is clear.

Lines 51-53: ,, We fixed a threshold of 7° to indicate an improvement in passive SLR angle and differentiate between 'Responders' and 'Non-responders'. “

Why did you choose this value of 7°?

Lines 53-55: ,,Additionally, we performed a correlation analysis between clinical characteristics and the improvement in the passive SLR angle in Responders, Non-responders, and all participants.”

What clinical features are you referring to?

Lines 79-83: ,,A sensitivity analysis considering a threshold of 6° to 8° of improvement in the passive SLR angle to differentiate between Responders and Non-responders revealed very similar distributions between the groups. Specifically, using a 6° threshold identified 33 Responders and 57 Non-responders, while a 8° threshold identified 25 Responders and 65 Non-responders”.

So what is the threshold? 7áµ’ as you state above? 6áµ’ or 8áµ’?

Lines 93-98:,, We performed this supplementary analysis to identify clinical differences between Responders and Non-Responders to passive HM stretching, however only BMI differed between groups; it was significantly smaller in Responders. The reason for the effect of BMI on stretch efficacy is not obvious. “

 Can you correlate the data obtained by you with studies from the specialized literature?

You mentioned the hypothesis and the limits of the study. More participants are needed.

However, it was interesting to correlate these data obtained according to the gender of the participants and to see if these changes, in respondents or non-respondents, are different in female or male participants. On the other hand, how do these changes affect costs from an economic and medical point of view.

Can you correlate the data obtained by you with studies from the specialized literature?

You mentioned the hypothesis and the limits of the study. More participants are needed.

However, it was interesting to correlate these data obtained according to the gender of the participants and to see if these changes, in respondents or non-respondents, are different in female or male participants. On the other hand, how do these changes affect costs from an economic and medical point of view.

Further studies are certainly needed.

I think the conclusions must be improved.

My comments are only intended to make the paper better. Good luck! 

Author Response

Comments 1:

It is an interesting topic.

However, there are many aspects that need to be improved

Lines 30-31: ,, recovery.[1–3].

Lines 47-49: ,,A detailed description of the study methods, measurement, and intervention procedure was presented in a previous article.  [4].

The template for placing the reference number in the text is not respected , respectively [ ].

Response 1: Thank you. This is now corrected.

Comments 2: Lines 45-47: ,,The original study was approved by a local ethics committee (Comité de Protection des Personnes – Ouest 1, Identifier: 2020T2-01_RIPH2 HPS_2019-A03000-57) and informed consent was obtained from all participants.”

What is the age group of the study participants? In the table below, you specify the average age, but what are the age limits?

Response 2: Thank you for this pertinent remark. We have added this information in the text. Line 53: “The participants' ages ranged from 18 to 60 years.”

Comments 3: Lines 47-49: ,,A detailed description of the study methods, measurement, and intervention procedure was presented in a previous article.  [4].

I think it was better to write briefly some aspects of the study you mention.

The objective is clear.

Response 3: Thank you for pointing out this issue. We have provided a brief explanation of the study method.

Line 49: “Briefly, the SLR angle was measured with a digital inclinometer before and immediately after a one-minute session of passive hamstring stretching. Clinical characteristics (e.g., age, sex, body mass index, Oswestry Disability Index, Fear-Avoidance Beliefs Questionnaire, Hospital Anxiety and Depression Scale) were recorded at baseline.”  

Comments 4: Lines 51-53: ,, We fixed a threshold of 7° to indicate an improvement in passive SLR angle and differentiate between 'Responders' and 'Non-responders'. “

Why did you choose this value of 7°?

 Response 4: Thank you for this pertinent question. We have included a more detailed explanation in the text.

Line 53: “In the present supplementary analysis we fixed a threshold of 7° to indicate an improvement in passive SLR angle and differentiate between 'Responders' and 'Non-responders' to passive hamstring stretching. This threshold was set based on the mean improvement in SLR angle of the less flexible lower limb which was 7° and the previously estimated Minimal Detectable Change which ranged from 6.8° to 10.4° [4]. “

Comments 5: Lines 53-55: ,,Additionally, we performed a correlation analysis between clinical characteristics and the improvement in the passive SLR angle in Responders, Non-responders, and all participants.”

What clinical features are you referring to?

 Response 5: Thank you for this question. The complete list of clinical characteristics assessed at baseline and used in the comparison is provided in Table 1. We have added clarifications about the main clinical characteristics earlier in the text to ensure the reference is clear.

Line 50-53 : ” Clinical characteristics (e.g., age, sex, body mass index, Oswestry Disability Index, Fear-Avoidance Beliefs Questionnaire, Hospital Anxiety and Depression Scale) were recorded at baseline.”

Comments 6: Lines 79-83: ,,A sensitivity analysis considering a threshold of 6° to 8° of improvement in the passive SLR angle to differentiate between Responders and Non-responders revealed very similar distributions between the groups. Specifically, using a 6° threshold identified 33 Responders and 57 Non-responders, while a 8° threshold identified 25 Responders and 65 Non-responders”.

So what is the threshold? 7áµ’ as you state above? 6áµ’ or 8áµ’?

 Response 6: Thank you for drawing attention to the need to clarify this issue. Indeed, the threshold is 7°. The verification of a threshold between 6° and 8° was done in addition to ensure that choosing 7° was the right decision and to verify whether using a threshold other than 7° would yield different results.

We have added more precise explanations in the Methods and Results sections.

Line 71-77: “Additionally, a sensitivity analysis was performed to verify whether using a threshold other than 7° would yield different results and to ensure the robustness of our findings. Specifically, we examined whether using a threshold between 6° and 8° for improvement in the passive SLR angle would change the comparison results of clinical characteristics between 'Responders' and 'Non-responders.”

Line 91-93: “A supplementary verification of a threshold of 6° to 8° for improvement in the passive SLR angle revealed very similar distributions between the groups.”

Comments 7: Lines 93-98:,, We performed this supplementary analysis to identify clinical differences between Responders and Non-Responders to passive HM stretching, however only BMI differed between groups; it was significantly smaller in Responders. The reason for the effect of BMI on stretch efficacy is not obvious. “

 Can you correlate the data obtained by you with studies from the specialized literature?

Response 7: Thank you for this constructive suggestion. We acknowledge that the relationship between BMI and the effects of hamstring stretching in CLBP patients has not yet been studied, so we cannot directly correlate our results with similar studies. However, we have included findings from studies indicating a relationship between reduced muscle flexibility and higher BMI in older women, as well as between higher BMI and decreased hamstring flexibility in relation to low back pain.

Line 107-110: “To our knowledge, no other study has evaluated the relationship between BMI and HM stretching effects in people with CLBP. However, reduced muscle flexibility has been associated with higher BMI in older women [5], and low back pain has been linked to both higher BMI and decreased HM flexibility [6].”

Comment 8: You mentioned the hypothesis and the limits of the study. More participants are needed.

However, it was interesting to correlate these data obtained according to the gender of the participants and to see if these changes, in respondents or non-respondents, are different in female or male participants. On the other hand, how do these changes affect costs from an economic and medical point of view.

Further studies are certainly needed.

Response 8: We have checked if gender of the participants make difference in Responders and Non-responders. Specifically, there was no difference in gender distribution in both groups. It is presented in the Table 1. We have now added this issue in the discussion section.

 Line 127-129: “We also noticed no significant difference regarding gender between Responders and Non-responders, indicating that being male or female is independent of the hamstring stretching effect.”

Comments 9: I think the conclusions must be improved.

 My comments are only intended to make the paper better. Good luck!

Response 9:  Thank you very much for all of your constructive remarks. We hope you find our responses satisfactory and that our manuscript is improved. We have also revised our conclusion as outlined below.

Line 154-158: “In conclusion, the only clinical factor found to relate to a positive response to HM stretching was a low BMI. This indicates that people with overweight or obesity are more likely to present an unsatisfactory response to hamstring stretching. Other clinical factors, such as age, gender, pain and disability levels, fear-avoidance beliefs, and symptoms of anxiety and depression, appear to have no significant impact.”

Round 2

Reviewer 1 Report

Comments and Suggestions for Authors

The manuscript has been improved, I do not have any other comments.

Reviewer 2 Report

Comments and Suggestions for Authors

I saw that you took into account what you wrote. Completions were necessary to give the article fluency.

Good luck!